# Second Trimester Amniotic Fluid Angiotensinogen Levels Linked to Increased Fetal Birth Weight and Shorter Gestational Age in Term Pregnancies

**DOI:** 10.3390/life14020206

**Published:** 2024-01-31

**Authors:** Dionysios Vrachnis, Alexandros Fotiou, Aimilia Mantzou, Vasilios Pergialiotis, Panagiotis Antsaklis, George Valsamakis, Sofoklis Stavros, Nikolaos Machairiotis, Christos Iavazzo, Christina Kanaka-Gantenbein, George Mastorakos, Petros Drakakis, Nikolaos Vrachnis, Nikolaos Antonakopoulos

**Affiliations:** 1National and Kapodistrian University of Athens Medical School, 11527 Athens, Greece; dionisisvrachnis@gmail.com (D.V.); alexandrosfotiou92@gmail.com (A.F.); 2Aghia Sophia Children’s Hospital, National and Kapodistrian University of Athens, 11527 Athens, Greece; amantzou@med.uoa.gr (A.M.); chriskan@med.uoa.gr (C.K.-G.); 3First Department of Obstetrics and Gynecology, National and Kapodistrian University of Athens Medical School, Alexandra Hospital, 11527 Athens, Greece; pergialiotis.vp@gmail.com (V.P.); panosant@gmail.com (P.A.); 4Second Department of Obstetrics and Gynecology, National and Kapodistrian University of Athens Medical School, Aretaieion Hospital, 11527 Athens, Greece; gedvalsamakis@yahoo.com; 5Third Department of Obstetrics and Gynecology, National and Kapodistrian University of Athens Medical School, Attikon Hospital, 11527 Athens, Greece; sfstavrou@yahoo.com (S.S.); nikolaosmachairiotis@gmail.com (N.M.); pdrakakis@hotmail.com (P.D.);; 6Department of Gynecologic Oncology, Metaxa Memorial Cancer Hospital of Piraeus, 18537 Piraeus, Greece; christosiavazzo@gmail.com; 7Unit of Endocrinology, Diabetes Mellitus and Metabolism, National and Kapodistrian University of Athens Medical School, Aretaieion Hospital, 11527 Athens, Greece; mastorakg@gmail.com; 8Vascular Biology, Molecular and Clinical Sciences Research Institute, St George’s University of London, London SW17 0RE, UK; 9Department of Obstetrics and Gynecology, University Hospital of Patras, Medical School, University of Patras, 26504 Patra, Greece

**Keywords:** angiotensinogen, renin–angiotensin system (RAS), amniotic fluid, second trimester of pregnancy, third trimester of pregnancy, small for gestational age (SGA), large for gestational age (LGA), fetal growth, birth weight

## Abstract

Background: Despite the considerable progress made in recent years in fetal assessment, the etiology of fetal growth disturbances is not as yet well understood. In an effort to enhance our knowledge in this area, we investigated the associations of the amniotic fluid angiotensinogen of the renin–angiotensin system with fetal growth abnormalities. Methods: We collected amniotic fluid samples from 70 pregnant women who underwent amniocentesis during their early second trimester. Birth weight was documented upon delivery, after which the embryos corresponding to the respective amniotic fluid samples were categorized into three groups as follows: small for gestational age (SGA), appropriate for gestational age (AGA), and large for gestational age (LGA). Amniotic fluid angiotensinogen levels were determined by using ELISA kits. Results: Mean angiotensinogen values were 3885 ng/mL (range: 1625–5375 ng/mL), 4885 ng/mL (range: 1580–8460 ng/mL), and 4670 ng/mL (range: 1995–7250 ng/mL) in the SGA, LGA, and AGA fetuses, respectively. The concentrations in the three groups were not statistically significantly different. Although there were wide discrepancies between the mean values of the subgroups, the large confidence intervals in the three groups negatively affected the statistical analysis. However, multiple regression analysis revealed a statistically significant negative correlation between the angiotensinogen levels and gestational age and a statistically significant positive correlation between the birth weight and angiotensinogen levels. Discussion: Our findings suggest that fetal growth abnormalities did not correlate with differences in the amniotic fluid levels of angiotensinogen in early second trimester pregnancies. However, increased angiotensinogen levels were found to be consistent with a smaller gestational age at birth and increased BMI of neonates.

## 1. Introduction

The fetal growth centile estimated at 24–42 weeks is based on birth weight and can vary depending on the ethnicity and geographic region of the population under study. The fetal growth curves used today have therefore been created by utilizing normative data from diverse populations and geographic regions. The rate of fetal growth velocity can be estimated through the use of sequential sonographic measurements of specific fetal body parts prior to delivery [1]. Small for gestational age (SGA) infants are neonates whose birth weight is less than the 10th percentile for gestational age [2]. Macrosomia is used to describe a large fetus or newborn, while large for gestational age (LGA) is the term used for fetuses or newborns with a weight above the 90th percentile for gestational age [3]. Appropriate for gestational age (AGA) are fetuses between the 10th and the 90th percentile.

However, not all fetuses classified as SGA or LGA are growth-restricted once maternal ethnicity, parity, maternal body mass index, and/or other parameters have been considered [3]. Notably, approximately 70% of SGA fetuses have normal perinatal outcomes despite their small weight at birth [4]. On the other hand, FGR (fetal growth-restricted) fetuses, that is, those that do not attain their complete growth potential due to pathological fetal or maternal factors, are more frequently correlated with unfavorable perinatal outcomes. The mortality rate of FGR was estimated to be from 1 to 6%, while those with AGA have a lower mortality rate of 0.2% [5,6].

FGR impacts a number of SGA fetuses, affecting approximately 3 to 6% of all deliveries [7,8]. A fairly extensive amount of data point to the possibility that an adverse prenatal environment and impaired fetal growth may result in fetal programming that predisposes the fetus to such conditions as hypertension, diabetes, and cardiovascular disease later in life [9].

Fetuses that experience fetal growth disorders, i.e., either SGA or LGA, have an increased risk of perinatal morbidity and mortality. Meanwhile, severe SGA fetuses (below the 3rd centile) as well as FGR fetuses face the greatest risk of perinatal complications, such as perinatal demise, intraventricular hemorrhage, hypoxic ischemic encephalopathy, sepsis, or bronchopulmonary dysplasia [5]. LGA fetuses have a heightened risk of adverse outcomes during labor, such as shoulder dystocia and/or perinatal brachial plexus palsy [10]. Despite the significant progress that has been achieved in the field of fetal monitoring and high-risk pregnancy management, the precise mechanisms and the underlying cause(s) of impaired fetal growth are yet to be fully elucidated [11,12].

The renin–angiotensin system (RAS), a circulating endocrine system and homeostatic signaling pathway, controls blood pressure and regulates fluid balance. Every element of the RAS is present in the placenta as early as the 6th week of gestation. Regarding the placental expression of these components, they may potentially function separately from the systemic RAS, influencing various functions such as villous and extravillous cytotrophoblast proliferation, extravillous cytotrophoblast migration, invasion, and development of placental blood vessels [13]. Disturbance of the placental RAS and the equilibrium between the vasoactive peptides, which are components of the RAS, may influence placental blood circulation, resulting in an insufficiency of nutrient delivery, thus leading to SGA and FGR [14].

Several published studies have investigated the ability of several biomarkers to detect abnormalities in fetal growth, including angiotensinogen [15,16]. Angiotensinogen, an a2-globulin precursor of angiotensin, is a yin hormone produced in the fetal brain, heart, and blood vessels, liver, kidney, adipose tissue, and adrenal glands, which regulates blood pressure via vasoconstriction, one of the main functions of the RAS. The RAS, and particularly angiotensin II, play an important role in blood pressure regulation through the circulating fluid volume. A number of published studies, though presenting conflicting results, have highlighted the potential role of angiotensinogen in the development and establishment of preeclampsia [17,18,19,20], while a limited number of studies have sought to correlate this precursor hormone with fetal growth abnormalities with regard to its expression in the placenta [21,22]. Furthermore, there is no literature on the amniotic levels of angiotensinogen among these pregnancies in any trimester. Of interest, published research shows that polymorphisms in the angiotensinogen gene in both maternal and fetal DNA could be associated with an increased risk of adverse pregnancy outcomes, including preeclampsia and intrauterine FGR [23].

The aim of our prospective observational study is to examine angiotensinogen in the amniotic fluid of early second trimester gestations, to investigate any correlations among groups, and to study its role in fetal growth abnormalities.

## 2. Material and Methods

This observational prospective cohort study included pregnant women who underwent amniocentesis during the early second trimester (17–21 weeks of gestation): the diagnostic test was performed due to such indications as previous history of birth defects, increased nuchal translucency, advanced maternal age, or detection of an anatomical anomaly in the ultrasound examination of the first or second trimester. The exclusion criteria were as follows: pregnancies with chromosomal abnormalities or major abnormalities diagnosed by fetal karyotype, in vitro fertilization pregnancies, and multiple pregnancies. All pregnant women who were diagnosed with hypertensive disorders (pre-existing or gestational) or diabetes (pre-existing or gestational) were excluded from our cohort population. Cases affected by such pathologies are known to be at high risk of fetal growth disturbances and are monitored closely during the pregnancy. Greater scientific lack of knowledge surrounds other groups of unpredictable cases with fetal growth disturbances who may suffer an unexpected unfavorable outcome. For this reason, our investigation focuses on the correlation between angiotensinogen’s amniotic fluid levels and fetal growth disturbances in an otherwise healthy pregnant population.

Gestational age was calculated based on the crown–rump length (CRL) measurement at the first trimester ultrasound, between the 11th and 14th week of gestation [24]. Follow-up was carried out until labor. Birth weight was documented at the time of delivery and thus the neonates were characterized as SGA, AGA, and LGA. Birth weight reflects the growth velocity in utero during the third trimester, so the same labeling can be attributed to the fetuses prior to delivery and be given to the amniotic fluid samples obtained in the second trimester. Thus, the samples were also categorized into three groups, namely, SGA, AGA, and LGA, meaning that the SGA samples are derived from the fetuses proved to be SGA as neonates at birth, the AGA samples are derived from the fetuses proved to be AGA as neonates at birth, and the LGA samples are derived from the fetuses proved to be LGA neonates at birth.

It was anticipated that some SGA fetuses would have FGR, while a number of LGA fetuses might exhibit severe macrosomia, which could result in early delivery, whether spontaneous or medically induced [1]. 

After amniocentesis, the collected samples underwent centrifugation, and the resulting supernatants were subsequently preserved in polypropylene tubes at a temperature of −80 °C until all samples were collected and analyzed. Amniotic fluid angiotensinogen concentrations were measured using the Human Angiotensinogen ELISA Kit (Life Technologies Corporation, Thermo Fisher Scientific Inc., Carlsbad, CA, USA) according to the manufacturer’s instructions. This concerns a solid-phase sandwich ELISA, which is designed to detect and quantify angiotensinogen levels in cell culture supernatants, plasma, and serum. The sensitivity of lower calibrators was evaluated, and the minimum detachable dose of human angiotensinogen was observed to be 1.22, based on the manufacturer’s product information sheet.

Statistical analysis was conducted using the Statistical Package for Social Sciences (SPSS) version 21 (IBM Corp., Armonk, NY, USA; Released 2021. IBM SPSS Statistics for Windows, Version 21), with the choice between parametric and nonparametric methods being dependent on the specific circumstances or requirements [25]. The distribution of the sample values was evaluated by regression analysis (Kolmogorov–Smirnov test). The results include the presentation of the median and interquartile range for quantitative variables. As per our study design, confounding factors taken into account include maternal age, body mass index, pregnancy duration, fetal sex, and multiparity. We established the threshold for statistical significance at a p-value lower than 0.05.

Approval by the Ethical Committee for Research of the Aretaieion University Hospital, Athens, Greece, was obtained with reference number 542-1310202. Moreover, all women participating in this study were informed, and a signed informed consent was obtained.

Table 1 presents the descriptive characteristics of both maternal and fetal aspects. There was no statistically significant difference observed among the three groups in terms of maternal age, maternal weight, and maternal height. However, statistical analysis found a significantly higher percentage of cesarean sections in the SGA group. As expected, statistically significant differences were observed between the groups in terms of gestational age at birth, birth weight, and centile.

## 3. Results

In this study, we assessed the concentrations of angiotensinogen present in the amniotic fluid of pregnancies undergoing amniocentesis in the early second trimester. In total, the angiotensinogen levels of 70 amniotic fluid samples were measured. Figure 1 represents the angiotensinogen levels for all subgroups. 

More specifically, the AGA subgroup’s mean angiotensinogen levels were found to be 4670 ng/mL (values from 1995 to 7250 ng/mL). Respectively, the SGA group’s mean level was 3885 ng/mL (values from 1625 to 5375 ng/mL), while the LGA group’s mean level was 4885 ng/mL (values from 1580 to 8460 ng/mL). Differences among the values of these three groups were not statistically significant (*p* = 0.676). Post-hoc analysis among these three groups did not find any statistically significant difference (Table 2).

Furthermore, we conducted multiple logistic regression analysis on independent parameters such as maternal age, maternal weight and height, gestational age, fetal birth weight, and percentile of birth weight. Table 3 summarizes the results of this multiple regression analysis. The multiple regression analysis revealed that increased angiotensinogen levels were correlated with lower gestational age at birth. Moreover, the same analysis highlighted the fact that the angiotensinogen levels increased in proportion to the fetal birth weight increase.

## 4. Discussion

Our study included pregnant women who underwent amniocentesis in the first half of the second trimester and were closely monitored until labor. While several amniotic fluid hormones or endocrine disruptors have previously been investigated in order to determine whether there is a possible correlation with fetal growth disturbances [26,27], the primary objective of our study was to determine whether there was potentially any correlation between the levels of angiotensinogen in the amniotic fluid and fetal growth disturbances.

Other authors have suggested the possible role of the placental RAS in the maturation and function of the placenta [28,29,30,31,32]. Although the extent of the RAS component expression has been investigated in placentas affected by FGR, either idiopathic or due to preeclampsia, there has, to our knowledge, been no investigation of angiotensinogen in the amniotic fluid. This research work is, as far as we know, the first to investigate the possible role of the RAS, and specifically of angiotensinogen, in the amniotic fluid of pregnancies with abnormal growth velocity in pregnancy. Our findings imply that angiotensinogen might play a role in compromising fetal blood circulation in cases of idiopathic SGA and FGR, as is shown below.

Our analysis did not reveal any statistically significant difference between the levels of angiotensinogen in the three subgroups. However, the statistical analysis reveals wide discrepancies in the mean values of angiotensinogen among the SGA, AGA, and LGA subgroups. More specifically, these differences reached a mean value of approximately 1000 ng/mL. Nevertheless, the large confidence intervals that were observed and were used for our statistical analysis resulted in a non-statistically significant difference among these groups, indicating that this hormone cannot be used as an amniotic fluid characteristic biomarker with high diagnostic accuracy. This finding has also been reported in other studies [33,34]. As demonstrated in our study and other previous ones, the larger the width of the confidential interval, the lower the diagnostic accuracy of the substance studied.

Despite the fact that our analysis did not find any statistically significant differences among the three subgroups, multiple regression analysis revealed a significant positive correlation between the levels of amniotic fluid angiotensinogen and birth weight. These highly interesting findings are in accordance with published research studies in adults. More specifically, a study in an African population by Cooper et al. found that the higher the BMI was, the higher the levels of serum angiotensinogen were [35]. A study by Engeli et al. reported that not only were the angiotensinogen levels in obese menopausal women higher than those of women with normal weight, but also that women who achieved a 5% reduction in their weight were observed to have 27% lower serum angiotensinogen levels than they had had previously (*p* < 0.05) [36].

Additionally, our multiple regression analysis revealed a negative correlation between the gestational age and amniotic fluid angiotensinogen levels. Similar results have been shown in the aforementioned study by Cooper et al. More specifically, the researchers revealed a negative correlation of angiotensinogen levels beyond middle age (over 34 years in women and 44 in men) [35]. These results are similar to our findings regarding the correlation between angiotensinogen levels and gestational age, third trimester in utero life, which, in fact, resemble late middle age and late adulthood when angiotensinogen levels drop.

Increased angiotensinogen could comprise one of several biological pathways leading to earlier delivery in cases with reduced growth velocity and may also contribute to advanced birth weight in other cases where the fetal weight is above the 50th centile. Thus, the correlation with the birth weight centile is ultimately minor. This could explain why there is no association between angiotensinogen and fetal growth extremes, which define the SGA and LGA groups.

Several researchers have investigated the role of the RAS in fetal growth. A recently published article investigated the role of the RAS in pregnancies complicated by FGR. The latter study reported a positive correlation between the levels of the m-RNA expression of angiotensinogen and birth weight in AGA fetuses; the authors further report that this correlation was not found in pregnancies with FGR, indicating that a lack of placental angiotensinogen expression could lead to FGR [37]. In our study, the angiotensinogen levels appeared to be lower in the amniotic fluid of SGA pregnancies: it is thus clear that future studies are required to investigate whether the m-RNA expression of the angiotensinogen gene is deficient in low-birth-weight pregnancies.

There is, moreover, evidence in the literature that some polymorphisms in the angiotensinogen gene such as AGT M235T could be associated with fetal growth disturbances, including SGA and FGR [38,39]. Another study found a statistically significant higher prevalence of this gene mutation in idiopathic FGR pregnancies compared to control pregnancies in a cohort population in Utah, USA [40]. Moreover, women carrying this allele gene were found to have higher levels of serum angiotensinogen even during the postpartum period [41]. In addition, other researchers concluded that fetal AGT M235T polymorphism was associated with low birth weight [42]. Our study did not aim to analyze these hypotheses; however, we anticipate that future research carried out in our cohort or a similar SGA population can investigate their incidence and their correlation with fetal growth disturbances.

The long-term effects of SGA and FGR on adult life have been extensively investigated [11,43,44,45]. A recently published study has explored the correlation between FGR, including SGA cases and preterm birth, and kidney size and kidney function by measuring several biochemical markers, such as the renin–angiotensin–aldosterone system (RAAS), in adolescents. Although the researchers observed that FGR and preterm birth were associated with smaller total kidney volume, no statistical difference in the biochemical markers of kidney function or RAAS components, including angiotensinogen, was found in this study, which is in accordance with our own angiotensinogen findings [46]. Moreover, these findings were consistent with the previous results of research into fetal growth-restricted sheep wherein the researchers reported significant reductions in the sheep’s kidney weight but no evidence of alteration in the renal RAS components [47]. All the above data revealed that fetal growth disturbances had no impact on the renal angiotensinogen levels in adolescents or adults who had experienced fetal growth disturbances at birth. 

It should be pointed out that collecting amniotic fluid is a particularly challenging procedure given the considerable difficulty of dealing with this biological material and assembling the needed number of cases. Therefore, one limitation of the current study is the small number of included cases, which consequently leads to a small sample size within the study subgroups. As far as we know, this is the first examination of amniotic fluid angiotensinogen levels during the second trimester. Previous researchers have investigated only the presence of the angiotensinogen gene or its expression in the placenta of FGR fetuses. This emphasizes the necessity for more extensive studies on amniotic fluid so as to clarify potential links between fetal growth irregularities and the different expression and mode of action of the RAS. It is hypothesized that many other factors influence the renin–angiotensin system and its numerous interactions in human and fetal physiology. Hence, it is hoped that our study may trigger further investigation in this direction.

## 5. Conclusions

This is the first study, to our knowledge, to investigate the presence of and possible correlation between angiotensinogen and fetal growth disorders in the amniotic fluid of early second trimester pregnancies. Our findings did not reveal any correlation between the levels of angiotensinogen in the amniotic fluid and, thus, in fetal growth disturbances. However, there were large discrepancies between the mean values of angiotensinogen in the three subgroups, a highly interesting finding that can be employed as the basis for further research. The mean value of SGA differs greatly from the AGA and LGA values. Moreover, multiple regression analysis identified a statistically significant correlation between the amniotic fluid angiotensinogen levels, gestational age, and birth weight. There are strong indications that angiotensinogen could constitute one of several biological pathways leading to earlier delivery in some cases and may also contribute to advanced birth weight in other cases. These correlations highlight a possible dual role of an RAS component in fetal growth velocity.

## Figures and Tables

**Figure 1 life-14-00206-f001:**
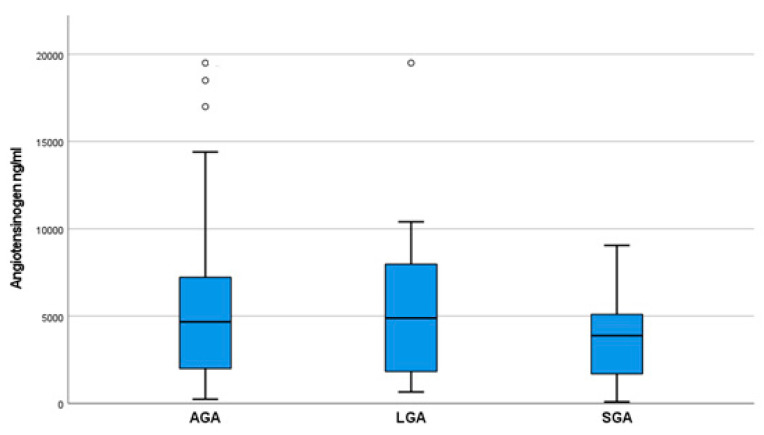
Angiotensinogen levels in AGA, LGA, and SGA groups. Box and whisker plots indicate box limits: Q1 and Q3. The dots in the upper part of the figure represent samples that brought back out-of-range increased measurements, located above the standard deviation of the mean value.

**Table 1 life-14-00206-t001:** Maternal and fetal characteristics.

	AGA [Mean (Upper and Lower Extreme)]	SGA [Mean (Upper and Lower Extreme)]	LGA [Mean (Upper and Lower Extreme)]	*p*-Value
**Maternal age (years)**	37 (28–43)	36 (26–41)	35 (29–43)	0.060
**Maternal weight (Kgr)**	60 (48–93)	62 (47–100)	59 (49–105)	0.650
**Maternal height (cm)**	165 (156–174)	167 (150–174)	166 (160–174)	0.070
**Gestational age (days)**	275 (261–285)	267 (261–283)	274 (268–282)	**0.011**
**Neonatal birth weight (gr)**	3290 (2860–3750)	2630 (1750–2860)	3800 (3550–4330)	**<0.001**
**Neonatal sex (female/all)**	19/33	3/18	10/19	**<0.001**

**Table 2 life-14-00206-t002:** Two-sided test. Each row tests the null hypothesis that the values of two groups are the same. Asymptotic significances (two-sided tests) are displayed. The significance level is 0.050.

Mean Values Compared between Groups	Significance (Two-Sided Test)
SGA–AGA	0.441
LGA–AGA	0.889
SGA–LGA	0.429

**Table 3 life-14-00206-t003:** Multiple regression analysis of independent parameters.

	Angiotensinogen	Age	Weight	Gestational Age	Birth Weight	Percentile
Angiotensinogen	1	0.208	0.139	−0.420 *	0.950 **	0.148
Age		1	0.035	0.76	0.148	0.088
Weight			1	−0.152	0.118	0.139
Gestational age				1	−0.157	−0.420 *
Birth weight					1	0.950
Percentile						1

* Correlation is significant at a level <0.05, ** Correlation is significant at a level <0.001.

## Data Availability

The data presented in this study are available on request from the corresponding author.

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
