# Peer review of "Second Trimester Amniotic Fluid Angiotensinogen Levels Linked to Increased Fetal Birth Weight and Shorter Gestational Age in Term Pregnancies"

_life, 2024, doi:10.3390/life14020206_

Round 1

Reviewer 1 Report

Comments and Suggestions for Authors

1. You have not mentioned anything about pregnancies associated with diabetes, collagen diseases, etc. from the perspective of placental vascular damage (level of angiotensin in the amniotic fluid). Were they excluded from the study? Also, those with pre-existing hypertension? (No patient was diagnosed with gestational hypertension, preeclampsia, or eclampsia)

2. Have you considered other factors that can influence the level of angiotensin in the amniotic fluid? maybe some discussions would be necessary……..

3. Does the level of angiotensin in the amniotic fluid vary according to the gestational age? can this aspect influence your results regarding fetal growth?

4. Interesting article…

Author Response

Dear reviewer,

Thank you for reviewing our article and your valuable remarks. All recommendations were taken into account and your questions have been addressed. Please find below our responses in detail. Furthermore, a revised version of the manuscript with the respective changes has been uploaded on the journal’s submission system. We hope that our article in its revised form meets the Journal’s criteria for acceptance and publication.

Sincerely,

Nikοlaos Antonakopoulos

Ass. Professor of Obstetrics and Gynecology

Reviewer 1

1. You have not mentioned anything about pregnancies associated with diabetes, collagen diseases, etc. from the perspective of placental vascular damage (level of angiotensin in the amniotic fluid). Were they excluded from the study? Also, those with pre-existing hypertension? (No patient was diagnosed with gestational hypertension, preeclampsia, or eclampsia)

Authors reply: Thank you for your comment. All cases with diabetes (pre-existing or gestational), collagen disease or hypertensive disorder (pre-existing or preeclampsia/eclampsia) were excluded from our study. Our aim was to investigate the role of angiotensinogen in the general population and, potentially, evaluate its performance as a screening tool in the future in the absence of such pathologies. We present our research results without being influenced by concomitant gestational conditions. Cases affected by the above pathologies are known to be at high risk for fetal growth disturbances. The present-day scientific lack of knowledge is perpetuated by cases with fetal growth disturbances without other pathology, who may eventually suffer an unexpected unfavorable outcome. Our investigation focuses on the correlation between angiotensinogen’s amniotic fluid levels and fetal growth disturbances in an otherwise healthy pregnant population. Based on your comment, the respective sentence in the “Material and Methods” section is now changed to:

All pregnant women who were diagnosed with hypertensive disorders (pre-existing or gestational) or diabetes (pre-existing or gestational) were excluded from our cohort population. Cases affected by such pathologies are known to be at high risk for fetal growth disturbances and are monitored closely during the pregnancy. Greater scientific lack of knowledge surrounds other groups of unpredictable cases with fetal growth disturbances who may suffer an unexpected unfavorable outcome. For this reason, our investigation focuses on the correlation between angiotensinogen’s amniotic fluid levels and fetal growth disturbances in an otherwise healthy pregnant population.

2. Have you considered other factors that can influence the level of angiotensin in the amniotic fluid? Maybe some discussion would be necessary…..

Authors’ reply: Thank you for your apt comment. As highlighted in our manuscript, there are no published data on amniotic fluid angiotensinogen in any gestational trimester. Our study was the first, as far as we know, designed to detect angiotensinogen’s presence and the possible concentration differences between different fetal growth velocities. However, after our favorable primary results, we conducted a multivariate analysis including some of the most common factors that could be linked to angiotensinogen levels. Furthermore, there are many other factors that could be investigated in the future, such as the renin-angiotensin system and its numerous interactions in human and fetal physiology.

Based on your comment the following sentence was added in the “Discussion” section:

It is hypothesized that many other factors influence the renin-angiotensin system and its numerous interactions in human and fetal physiology. Hence, it is hoped that our study may trigger further investigation in this direction.

3. Does the level of angiotensin in the amniotic fluid vary according to the gestational age? Can this aspect influence your results regarding the fetal growth?

Authors’ reply: Thank you for your comment. The majority of our samples were obtained within the same gestational window, i.e., the recommended gestational period for amniocentesis between 17 and 21 weeks, thus a significant effect of the exact gestational week is not expected, taking further into account the random distribution of cases. The time period of amniocentesis is now added in the “Material and Methods” section of the manuscript. On the other hand, our multiple regression analysis revealed a negative correlation between gestational age at birth and amniotic fluid angiotensinogen levels. This finding is highlighted and further discussed in our “Discussion” section, where a parallel is drawn with adult life, as reported in the published literature.

  1. Interesting article

Authors’ reply: Thank you for your kind comment. Our research team has made a significant effort over the last decade in order to shed light upon the complex pathways of fetal growth and thereby develop possible explanations and etiologies for several gestational pathologies that are closely associated with perinatal morbidity and mortality rates, such as fetal growth restriction, fetal macrosomia, and preterm labor. This recognition from experts in the field is very motivational for us and highly appreciated.

Reviewer 2 Report

Comments and Suggestions for Authors

The study attempted to tackle an interesting topic and find some clinically relevant marker for prediction of fetal growth abnormality. However, the manuscript contains a few questionable points which should be addressed before considering it for publication:

1.      The Title does not reflect the results of the study. The main goal of the study was to find the link between amniotic fluid angiotensinogen and fetal growth abnormalities. So, the title can be re-written to be more close to major finding “fetal growth abnormalities did not correlate with differences in the amniotic fluid levels of angiotensinogen”.

2.      Introduction and Discussion are too long for amount of obtained data. The text could be shortened by omitting information not directly relevant to the subject of the work.

3.      Keywords: fetal growth restriction (FGR) was not evaluated in the study.

4.      Lines 31 and 122-123: the sentence is awkward, I suppose the amniotic fluid samples have been categorized according to newborns weight. By the way, why the term “embryos” is used for newborns?

5.      Table 1: please, check neonatal birth weight SGA 3800 (3550-4330) and LGA 2630 (1750-2860).

6.      Table 2 is not necessary, an absence of statistical significance is clearly seen on the graph.

7.      Figure 1: please, explain the meaning of digits at the dots in the legend to the figure. Next, for what reason the values at Y axis are presented as thousands of ng/ml, while in the text (page 4, lines 166-168) IU/ml are used? Please, unify the units and present the same ones in Abstract, main text and on the graph.

8.      There is no Table 3 in the work.

9.      Some conclusions are quite speculative, for example, that presented at the lines 217-223. The comparison of fetal third trimester (rapid growth and development, high proliferation and differentiation in all tissues and organs) and late adulthood (senescence, degenerative processes) is not appropriate.

10.  Lines 229-230: the statement contradicts experimental findings – angiotensinogen level was not lower in amniotic fluid of SGA pregnancies. In fact, the statement ”increased angiotensinogen levels were found to be consistent with smaller gestational age at birth and increased BMI of neonates” is somewhat strange, like LGA newborns were born at earlier gestational age.

11.  There are some stylistic errors in the text. For example, line 133: “this concerns a solid-phase sandwich…”

Comments on the Quality of English Language

Minor English editing is required.

Author Response

Dear reviewer,

Thank you for reviewing our article and your valuable remarks. All recommendations were taken into account and your questions have been addressed. Please find below our responses in detail. Furthermore, a revised version of the manuscript with the respective changes has been uploaded on the journal’s submission system. We hope that our article in its revised form meets the Journal’s criteria for acceptance and publication.

Sincerely,

Nikοlaos Antonakopoulos

Ass. Professor of Obstetrics and Gynecology

Reviewer 2

The study attempted to tackle an interesting topic and find some clinically relevant marker for prediction of fetal growth abnormality. However, the manuscript contains a few questionable points which should be addressed before considering it for publication:

  1. The Title does not reflect the results of the study. The main goal of the study was to find the link between amniotic fluid angiotensinogen and fetal growth abnormalities. So the title can be re-written to be more close to major finding “fetal growth abnormalities did not correlate with differences in the amniotic fluid levels of angiotensinogen”

Authors’ reply: Thank you for your comment. Indeed, the initial aim of our research was to investigate the link between the fetal growth abnormalities and amniotic fluid angiotensinogen levels in the early second trimester of pregnancy. Although a trend was detected, this correlation was not found to be statistically significant. However, through our multiple regression analysis, two very interesting statistical correlations were revealed, namely, a link between birth weight and angiotensinogen levels and a link between gestational age at birth and angiotensinogen levels. Our findings imply that amniotic fluid angiotensinogen is linked to pathways leading to advanced gestational duration and fetal weight at birth. Birth weight is not the same as growth, which is reflected by fetal growth centiles. This explains why we have chosen as the article’s title “birth weight and gestational age”, which are both related to growth. We have thus particularly highlighted these correlations so that the reader can grasp from the very first moment what differs significantly. We request that the title be kept as it is.

  1. Introduction and Discussion are too long for amount of obtained data. The text could be shortened by omitting information not directly relevant to the subject of the work.

Authors’ reply: Thank you for your remark. The manuscript has been read, edited, and approved by all the contributing authors. None of us found the length of the article too long. We also sought to adhere strictly to the journal’s instructions to authors concerning our submitted manuscript’s length. Actually, we were asked by the Editor to increase the word count—this being considered necessary for the Editor prior his sending the article to you for review. As the article’s subject is original and it is the first time, to the best of our knowledge, that these data on angiotensinogen are being presented in the literature, we endeavored to provide a short but fairly comprehensive literature review in the introduction section, presenting all the existing background knowledge on the subject and explain the theory behind our hypothesis. The same goes for the discussion section, where we made an effort to analyze and explain our findings to enable the readers to have a clear view of the results and help other researchers to extend this investigation. This was judged important and useful for the readers given the results of the statistical analysis.

  1. Keywords: fetal growth restriction (FGR) was not evaluated in the study

Authors’ reply: Thank you for your comment. Indeed, our research did not evaluate directly FGR fetuses but SGA fetuses - despite the fact that most severe SGA fetuses are actually FGR. Thus, our findings partially reflect this group of fetuses, while we have also not used formal criteria to compare these fetuses with the remaining SGA fetuses or AGA controls. We have now removed the FGR keyword from our keywords list.

  1. Lines 31 and 122-123: the sentence is awkward; I suppose the amniotic fluid samples have been categorized according to newborns weight. By the way, why the term “embryos” is used for newborns?

Authors’ reply: Thank you for your comment. The amniotic fluid samples were categorized based on neonatal birth weight. We changed the sentence to render it more comprehensive, as follows:

Birth weight was documented at the time of delivery and, in accordance with the respective amniotic fluid samples, the samples were categorized into three groups, namely, SGA, AGA and LGA

The embryos of the early second trimester during which the amniocenteses were carried out developed in the late second and third trimester into SGA, LGA, and AGA embryos. The same birth weight corresponds concurrently to the embryo at the time of birth (before it breathes) and to the newborn (after it takes its first breath).

  1. Table 1: please check neonatal birth weight SGA 3800 (3550 – 4330) and LGA 2630 (1750 – 2860)

Authors’ reply: Thank you for your comment. We would like to apologize for this typing error. We have corrected the table as follows:

neonatal birth weight SGA 2630 (1750 – 2860) and LGA 3800 (3550 – 4330) 

  1. Table 2 is not necessary, an absence of statistical significance is clearly seen on the graph.

Authors’ reply: Thank you for your comment. Table 2 actually highlights the main results of our study concerning our initial research question. No statistically significant difference of angiotensinogen levels in the three studied subgroups (SGA, AGA, and LGA) was found. Although we agree that the graph implies a lack of statistical significance, it is however mandatory and common practice in all papers to present a table with the exact numbers of significance value. Some readers are not familiar with box and whisker plots graphs, so we feel that Table 2 makes the endpoint results of our study clear for all.

  1. Figure 1: please, explain the meaning of digits at the dots in the legends to the figure. Next, for what reason the values at Y axis are presented as thousands of ng/ml, while in the text (page 4, lines 166-168) IU/ml are used? Please, unify the units and present the same ones in Abstract, main text and on graph.

Authors’ reply: Thank you for your observation. The digits beside the dots in the upper part of Figure 1 are the numbers of the specific anonymous amniotic fluid samples, which brought back out-of-range increased measurements which are located far above the standard deviation of mean values. Concerning the measurement unit used for angiotensinogen, we apologize for the typing error. The units are now correctly written in the abstract. We have also changed the units accordingly in the whole manuscript, as they should appear at the time of our article submission (ng/ml).

  1. There is no Table 3 in the work.

Authors’ reply: Thank you for your comment. We apologize for the typing error due to an extra table that was removed from the submitted version of the manuscript. We have now changed the heading Table 4 to Table 3.

  1. Some conclusions are quite speculative, for example, that presented at the lines 217-223. The comparison of fetal third trimester (rapid growth and development, high proliferation and differentiation in all tissues and organs) and late adulthood (senescence, degenerative processes) is not appropriate.

Authors’ reply: Thank you for your comment. Our study is the first, to our knowledge, to investigate the presence and the possible correlation between angiotensinogen levels in amniotic fluid and fetal growth potential. We took on board your comment; the reference has remained but the comparison is now removed. This part of the article is thus now:

These results are similar to our findings regarding the correlation between angiotensinogen levels and gestational age, third trimester in utero life, which, in fact, resemble late middle age and late adulthood when angiotensinogen levels drop.”

  1. Lines 229-230: the statement contradicts experimental findings – angiotensinogen level was not lower in amniotic fluid of SGA pregnancies. In fact, the statement “increased angiotensinogen levels were found to be consistent with smaller gestational age at birth and increased BMI of neonates” is somewhat strange, like LGA newborns were born at earlier gestational age.

Authors’ reply: Thank you for your comment. This statement was our effort to highlight the fact that multiple regression analysis revealed a positive correlation between angiotensinogen levels and fetal birth weight, while a negative correlation was found between angiotensinogen levels and gestational age at birth. Thus, in accordance with your recommendation, we rephrased this sentence as:

Multiple regression analysis revealed that increased angiotensinogen levels were correlated with lower gestational age at birth. Moreover, the same analysis highlighted the fact that angiotensinogen levels increased in proportion to fetal growth weight increase.

Although these findings seem to be to a degree unforeseeable, they are not mutually contradictory. Angiotensinogen may comprise one component of several pathways leading to earlier delivery in cases with reduced growth velocity and may also contribute to advanced birth weight in other cases. Thus, the correlation with birth weight centile is found to be minor. This could explain why there is no association between angiotensinogen and fetal growth extremes that define the SGA and LGA groups. Increased fetal birth weight corresponds to all centiles above the 50th. The above hypothesis is now added in the discussion and reads:

Increased angiotensinogen could comprise one of several biological pathways leading to earlier delivery in cases with reduced growth velocity and may also contribute to advanced birth weight in other cases where the fetal weight is above the 50th centile. Thus, the correlation with birth weight centile is found to be minor. This could explain why there is no association between angiotensinogen and fetal growth extremes, which define the SGA and LGA groups.”

  1. There are some stylistic errors in the text. For example, line 133: “this concerns a solid-phase sandwich…”

Authors’ reply: Thank you for your comment. The phrase contained in this statement was taken from the manufacturer’s official leaflet attached to the ELISA kit used. Thus, we suggest that it be left as the manufacturer defines it.

Round 2

Reviewer 2 Report

Comments and Suggestions for Authors

Although the authors answered the questions raised during first round of review, there are still a few points which should be addressed:

1.      Lines 132-134: After changing the word “embryos” with the word “samples” the sentence became even more confusing since SGA, AGA and LGA are neonates but not samples. Although an entire sentence sounds awkward for me (probably because English is not my native language), the first version was better.

2.      Figure 1: the scaling of Y axis is thousands of ng/ml, while the values presented in the main text (lines 176-178) are within the range of ten. Please, correct.

3.      Next, I do not understand the meaning of circles with the digits on the graph (24, 47 and 32 above AGA plot and 67 above LGA plot). Please, add the detailed description to the legend to figure.

Author Response

Dear reviewer,

Thank you for reviewing again our article and giving your valuable feedback. All recommendations were taken into account and your questions have been addressed. Please find below our responses in detail. Furthermore, a revised version of the manuscript with the respective changes has been uploaded on the journal’s submission system. We hope that our article in its revised form meets the Journal’s criteria for acceptance and publication.

Sincerely,

Nikοlaos Antonakopoulos

Ass. Professor of Obstetrics and Gynecology

Reviewer 2

Although the authors answered the questions raised during first round of review, there are still a few points which should be addressed:

  1. Lines 132-134: After changing the word “embryos” with the word “samples” the sentence became even more confusing since SGA, AGA and LGA are neonates but not samples. Although an entire sentence sounds awkward for me (probably because English is not my native language), the first version was better.

Authors reply: Thank you for your comment. The samples were categorized into three groups, namely, SGA, AGA, and LGA, meaning that SGA samples are derived by fetuses proved to be SGA as neonates at birth, AGA samples are derived by fetuses proved to be AGA as neonates at birth and LGA samples are derived by fetuses proved to be LGA neonates at birth. The entire sentence was revised to be more clear as follows: “Birth weight was documented at the time of delivery and thus the neonates were characterized as SGA, AGA and LGA. Birth weight reflects the growth velocity in utero during the third trimester, so the same labeling can be attributed to the fetuses prior delivery and be given at the amniotic fluid samples obtained in the second trimester. Thus, the samples were categorized also into three groups, namely, SGA, AGA, and LGA, meaning that SGA samples are derived by fetuses proved to be SGA as neonates at birth, AGA samples are derived by fetuses proved to be AGA as neonates at birth and LGA samples are derived by fetuses proved to be LGA neonates at birth.”.

  1. Figure 1: the scaling of Y axis is thousands of ng/ml, while the values presented in the main text (lines 176-178) are within the range of ten. Please, correct.

Authors reply: Thank you for your comment. We apologize for the typing error. In the abstract section it is written correctly. Thus, this is now changed in the manuscript main text as well.

  1. Next, I do not understand the meaning of circles with the digits on the graph (24, 47 and 32 above AGA plot and 67 above LGA plot). Please, add the detailed description to the legend to figure.

Thank you for your observation. The digits beside the dots in the upper part of Figure 1 are random numbers given by the lab to the specific amniotic fluid samples, which brought back out-of-range increased measurements (thus located above the standard deviation of mean values). We have now erased these numbers, while the dots were left on the graph. The description to the legend of the figure now reads: “The dots in the upper part of the figure represent samples which brought back out-of-range increased measurements, located above the standard deviation of the mean value”.